# Fast food over safe food? A qualitative evaluation of a food safety training intervention for street vendors applying the COM-B model in Ouagadougou, Burkina Faso

**Dónya S. Madjdian**[1]*, **Vera Dankwah Badu**[1], **Guy Ilboudo**[2], **Valerie R. Lallogo**[2], **Michel Dione**[2], **Marcel van Asseldonk**[3], **Theodore J. D. Knight-Jones**[4], **Emely de Vet**[1,5]

1 Consumption and Healthy Lifestyles Group, Wageningen University & Research, Wageningen, The Netherlands, 2 International Livestock Research Institute (ILRI), Ouagadougou, Burkina Faso, 3 Wageningen Economic Research, Wageningen, The Netherlands, 4 International Livestock Research Institute (ILRI), Addis Ababa, Ethiopia, 5 University College Tilburg, Tilburg University, Tilburg, The Netherlands

* donya.madjdian@wur.nl

**Data Availability Statement:** Data cannot be shared publicly because of the risk of revealing

## Abstract

The safety of ready-to-eat food sold in urban informal markets in low and middle-income countries is a pressing public health challenge, that needs to be addressed if we are to establish healthy food systems. Guided by the Capability, Opportunity, and Motivation model for Behavior change (COM-B), this qualitative study aimed to explore perceptions of street vendors on their participation in a food safety capacity building intervention, consisting of training and provision of food safety equipment. The intervention aimed to improve food safety behavior of vendors of ready-to-eat chicken in informal markets in Ouagadougou, Burkina Faso. A total of 24 vendors selling ready-to-eat chicken at street restaurants participated in semi-structured interviews after training, which focused on vendors' stories of change related to food safety capabilities, opportunities, motivation, and behaviors. Data were thematically analyzed following COM-B components. Vendors noted improvements in psychological (i.e., knowledge, awareness, self-efficacy, perceptions) and physical capabilities (i.e., equipment useability and applicability), and motivations (perceived responsibility, reputation, client satisfaction, profits, consumer demand). Moreover, training and provision of equipment, spill-over effects to employees or neighboring outlets, and social support were perceived as key social and physical opportunities, while structural challenges such as market infrastructure, regulations, financial resources, cost of living, and outlet culture were physical barriers to implement lessons learnt. This study provides insights into the impact of engaging vendors in improving food safety behavior through training and equipment provision. Improvements in vendors' perceived capabilities and motivation contributed to improved food safety behavior, while contextual barriers hindered the perceived adoption of food safety behaviors.

vendor information considering the study's relatively small region. A data analysis coding table with codes, categories, themes and example quotes is available as Supporting Information. Full transcripts are available upon request from the International Livestock Research Institute (ILRI) via MELDATA Dataverse (https://data.mel.cgiar.org/).

**Funding:** This study was financially supported by the Bill & Melinda Gates Foundation [INV-008430-OPP1195588], UK Government Foreign, Commonwealth & Development Office (FCDO) - UK Aid from the United Kingdom government and the CGIAR Research Program on Agriculture for Nutrition and Health.

**Competing interests:** No potential conflict of interest was reported by the authors.

## Introduction

Sub-Saharan Africa's traditional informal markets are a key source of fresh, affordable, and easily accessible food [1]. However, because of limited regulations and infrastructure, poor environmental conditions, and unhygienic food handling practices by vendors, foods sold at these markets significantly contribute to foodborne diseases, thereby hampering food and nutrition security and exacerbating socio-economic inequalities [2,3]. Ready-to-eat animal-source food sold at local eateries, are a particular high-risk food category [4]. In Burkina Faso, flamed, braised, and grilled chicken are popular dishes among urban consumers [5]. Three-quarters of chicken meat is consumed at, or taken away from local eateries [6]. Compared to beef, dairy, and vegetables, the levels of microbial contamination in poultry meat and the resulting disease burden are high [7]. In Burkina Faso alone, the estimated annual economic losses attributed to foodborne disease were US$3 billion in 2018 [8] accounting for 3% of the gross national income per capita (GNI) [9]. *Escherichia coli* and *Salmonella spp*. prevalence in poultry meat ranged between 29 to 45%, and 57% respectively [8]. Approximately one out of 50 consumers fell sick, while one out of 30,000 persons in the total population died from consuming contaminated poultry meat in 2017 [7].

At the point of sale, unhygienic handling and preparation practices by vendors during slaughtering, scalding, plucking, washing, and grilling, contribute to high contamination levels in ready-to-eat chicken [6]. Contamination has been linked to poor personal and environmental hygiene, improper transporting and storage, unclean materials, and contaminated water [5]. Hence, vendors are key to preventing foodborne disease [4]. Evidence, however, points towards low food safety knowledge and awareness deficits resulting in high-risk practices among food handlers in the informal sector [1,10] For instance, a survey among poultry vendors in Ouagadougou, Burkina Faso, revealed poor food handling and preparation practices, despite awareness of food safety requirements [11]. Despite efforts to promote food safety in low and middle-income countries (LMICs), informal food value chains are still largely ignored by governments, with minimal on-ground efforts and initiatives that often overlook vendors [3]. Generally, interventions targeting vendors heavily rely on education to improve safe food handling [12], assuming that increased knowledge will change attitudes and behavior [13]. Several studies reported on the potential effectiveness of food safety training in enhancing knowledge and awareness of risky food practices [14,15], and overall effectiveness of training in improving vendor food safety behaviors in LMICs [13,15]. Other studies, however, observed that despite initial behavioral improvements, hygiene at outlets not always improved, and behavior change did not necessarily sustain over time due to physical barriers [16]. The challenge of on-off training programs without ongoing support systems could be attributed to reliance on traditional knowledge-sharing models, and discrepancies with knowledge-application contexts [17]. For behavior change, knowledge should be combined with attitudinal and motivational change, increased skills, and opportunities to implement the required behavior [18]. Thus far, research evaluating food safety interventions has generally focused on knowledge and behavior changes, with limited attention to vendors' capabilities, opportunities, and motivations that drive behavior change [19,20]. For example, evidence on motivational change after participating in interventions remains mostly anecdotal, with only a few studies aiming to understand motivations using qualitative techniques, while these are better suitable for understanding the underlying, hidden drivers or barriers behind actions [12,19].

This study therefore aimed to provide a comprehensive understanding of ready-to-eat chicken street vendors' capabilities, opportunities, and motivation for behavior change, after participating in a food safety training intervention at informal markets in Ouagadougou, Burkina Faso through qualitative research. The training intervention, implemented in October

and November 2022, aimed to improve vendors' food safety knowledge, attitudes, and practices, through participatory training and the distribution of enabling equipment. A quantitative baseline assessment, conducted as part of a Randomized Controlled Trial (RCT), which assessed the impact of this intervention through outlet observations and vendor surveys reported elsewhere [21], showed relatively poor food safety knowledge and practices prior to the training. For example, on a ten-point knowledge score, mean knowledge score was six, pre-training. In terms of chicken carcass management and chicken preparation, less than half of the vendors regularly renewed water when washing raw chicken carcasses or handled prepared chicken while wearing gloves or using a fork and knife. Less than a quarter of outlets had an adequate handwashing facility for employees and customers, and more than half of the outlets were observed to have poor waste management practices [22]. Additionally, a previous Knowledge-Attitudes-Practices survey—part of the same research project conducted among 100 vendors in Ouagadougou—showed that only one out of five food handlers had implemented strategies to improve food safety at their outlet, while four out of five food handlers believed that cleanliness and hygiene were not important to their customers [23].

To gain a deeper understanding of the impact of the training on the by vendors' perceived capabilities, opportunities, and motivations related to food safety behaviors following their participation, and to further inform the sustainable scale-up of effective food safety vendor training interventions in Sub-Saharan Africa, we conducted a qualitative study in parallel with the RCT (reported elsewhere [22], with the vendors who were part of RCT's treatment arm.

## Methods

We conducted a total of 24 qualitative in-depth interviews complemented with the Most Significant Change technique, with 24 vendors from the RCT's treatment arm. The study was guided by the Capability-Opportunity-Motivation-Behavior (COM-B) model, which identifies three interacting components influencing one's behavior: capability, opportunity, and motivation [24]. Capability involves psychological capabilities including cognitive skills and knowledge, and physical skills and competencies, that enable vendors to safely handle food. Vendors must understand safe food practices and correctly use equipment to handle chicken meat safely, whilst risk perceptions, shaped by knowledge and experience, may influence adoption [25]. Opportunity encompasses external factors that facilitate or hinder behavior, including the sociocultural and physical environment [24]. Social opportunities involve support or norms, while economic opportunities include financial resources and access to basic needs like potable water, sanitation, and waste management, which street vendors often lack [26]. Motivation can be reflective and automatic [24]. Reflective motivation involves conscious decision-making, encompassing deliberative goal-setting and intentions based on beliefs and values, while automatic motivation refers to impulsive influences such as emotions, habits, and immediate responses without conscious awareness [27]. Optimal behavior change is achieved when all three components are effectively aligned and supportive of the desired behavior, see Fig 1 [24].

### Training intervention

The training intervention aimed to reduce contamination of food through building the capacity of chicken value chain actors to cost-effectively alleviate important food safety risks in Burkina Faso [28,29]. The intervention targeted street vendors of ready-to-eat chicken in Ouagadougou, Burkina Faso, who are mostly men [23], and aimed to improve food safety knowledge, attitudes, and practices [30]. The intervention's design was based on insights from chicken value chain assessments, vendor knowledge, attitude, and practices surveys,

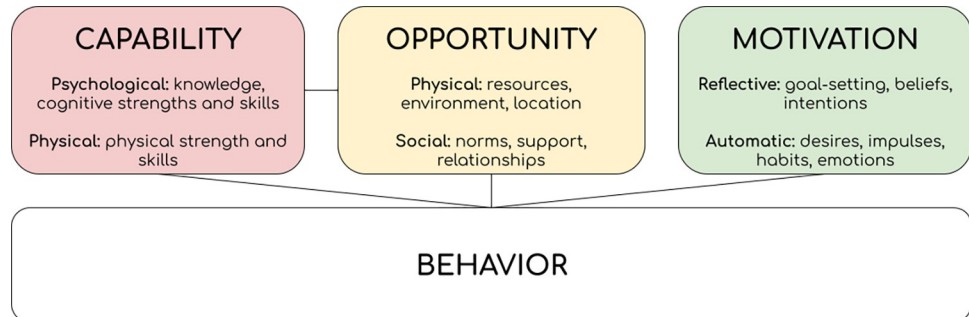

**Fig 1. COM-B model (adapted from [24]).**

stakeholder engagement meetings in Ouagadougou, and the WHO's Five Keys to Safer Food guidelines [11,31,32]. Learning outcomes were then contextualized and tailored to vendors' needs to support them to take ownership of everyday food safety management. The intervention consisted of three half-day participatory trainings, including nine interactive food safety modules for employees and outlet owners (S1 Table) delivered through lectures, quizzes, group discussions, demonstrations, and market visits, and a business skills module for owners, specifically [33]. Secondly, upon completion, participants received a low-cost, renewable, and easy-to-use equipment package including a jug and basin for handwashing, liquid soap, protective clothing, a cleaning sponge, cutting board, plastic tablecloth, aprons, and a garbage bin, as well as a certificate of attendance and good practices poster. The training was delivered by trained food safety regulators. A total of 72 outlets in the treatment group received training and equipment, with a maximum of vendors from 20 outlets attending one of the six training sessions, held between October and November 2022. A total of 90 outlets from the Control group received training and equipment after the study ended, between April and May 2023.

## Participants

We randomly sampled 24 vendors from the 72 outlets who were randomized into the treatment group (between 10 and 28 December 2022). To account for potential differences between outlet owners and employees with regards to the implementation of lessons learnt, we purposively sampled 12 outlet owners and 12 employees. Selected participants were contacted via phone, informed about the aim of the study, and invited to participate by the interviewers. Four sampled outlet owners and five employees declined the invitation due to reasons of being out of town, no time or attending to other obligations, or not being able to complete the full training. A second round of sampling resulted in reaching the desired sample of 24 interviews. In addition, we sampled an outlet employee and owner for the purpose of conducting pilot interviews.

## Data collection

Qualitative data were collected between end of December 2022 and February 2023. Two research assistants fluent in French and the local language *Mooré*, were trained by the researchers (DM, VDB) to conduct the interviews at vendors' outlets. Interviewers were not informed about the training content to avoid steering towards certain outcomes and guarantee neutrality [34]. Interviews were facilitated by a guideline (see S2 Table) which was translated to French as part of training. To build rapport, the guide began with questions about the interviewee's role and outlet information, followed by five unordered modules: food safety risk perceptions,

training content and objectives, training impact on practices, opportunities and barriers to food safety, and vendor-consumer food safety interactions.

Using the 'Most Significant Change' technique [35], a form of participatory monitoring and evaluation, we invited vendors to share the most significant change they experienced after the training through storytelling. When collecting these stories, the focus was given to 'what' change had occurred and 'why' vendors believed the change had happened [35]. After training and mock interviews, two pilot interviews (excluded from data analyses), were conducted, evaluated, and the guide was adjusted to resolve ambiguities and improve flow. Interviews were audio-recorded and lasted between 31 and 87 minutes. After each interview, interviewers answered questions to reflect on findings and note important observations.

## Analysis

Interview recordings were transcribed verbatim by interviewers to French, then translated into English by an independent translator. Audio recordings were cross-checked with both the original and translated transcripts for consistency and clarity. Transcripts were de-identified, with interviewees given pseudonyms for confidentiality. Data were analyzed using thematic analysis [36], integrating deductive themes based on the COM-B model, with inductive exploration of emerging themes. A preliminary coding scheme was developed. Two researchers (DM, VDB) independently coded all interviews and interviewer notes using Atlas Ti software after multiple readings. Themes, sub-themes, and codes were discussed, leading to merging or regrouping several before finalizing the coding scheme (S3 Table).

## Ethics

Ethical approval was obtained from the ILRI Institutional Research Ethics Committee (IREC, ILRI-IREC2021-63) and Comité d'éthique, Burkina Faso (CERS, 2022-11-232). All participants provided written consent and obtained permission from outlet owners if they were employees. They could withdraw or pause interviews at any time. Participants received a token of appreciation along with the equipment.

## Inclusivity in global research

Additional information regarding the ethical, cultural, and scientific considerations specific to inclusivity in global research is included in the (see S1 File).

## Results

### Participant characteristics

An overview of socio-demographic characteristics of the 12 outlet owners and 12 employees is presented in Table 1. Except for one, all participants were men. The average age of 24 interviewees was 34.5 years. Outlet size, the number of employees working at the time of the training, averaged four, with a minimum of one and maximum of 15 employees. Most vendors had not obtained any (formal) education, four vendors completed primary school, and three vendors attained secondary or higher education.

Fig 2 below summarizes the main results mapped onto the COM-B model, outlining prerequisites for food safety behavior changes.

### Capabilities

**Psychological capabilities.** *Knowledge and awareness*. Vendors reported that training significantly enhanced their knowledge and awareness of food safety. They specifically noted

**Table 1. Socio-economic characteristics of participants.**

| No | Name (pseudonym) | Sex | Role | Age | No. of employees at outlet | Educational attainment |
|---|---|---|---|---|---|---|
| 1 | Abdulai | Male | Employee | 25 | 2 | Secondary education |
| 2 | Abubakar | Male | Owner | 39 | 2 | No (formal) education |
| 3 | Ahmed | Male | Owner | 25 | 1 | No (formal) education |
| 4 | Alhaji | Male | Employee | 40 | 3 | No (formal) education |
| 5 | Amina | Female | Employee | 24 | 6 | Secondary education or higher |
| 6 | Awudu | Male | Owner | 34 | 3 | Primary education |
| 7 | Baba | Male | Owner | 28 | 2 | Secondary education |
| 8 | Bashiru | Male | Employee | 28 | 4 | No (formal) education |
| 9 | Christian | Male | Employee | 49 | 5 | No (formal) education |
| 10 | Fatawu | Male | Owner | 42 | 4 | Primary education |
| 11 | Hakim | Male | Owner | 28 | 1 | No (formal) education |
| 12 | Jacques | Male | Owner | 25 | 3 | No (formal) education |
| 13 | Jean | Male | Employee | 27 | 5 | No (formal) education |
| 14 | Ibrahim | Male | Owner | 36 | 5 | No (formal) education |
| 15 | Issah | Male | Employee | 23 | 5 | Primary education |
| 16 | Karim | Male | Employee | 26 | 4 | No (formal) education |
| 17 | Mohammed | Male | Employee | 36 | 3 | Primary education |
| 18 | Paul | Male | Employee | 33 | 4 | No (formal) education |
| 19 | Roland | Male | Owner | 48 | 15 | No (formal) education |
| 20 | Richard | Male | Employee | 37 | 2 | No (formal) education |
| 21 | Sherif | Male | Employee | 31 | 3 | No (formal) education |
| 22 | Sulemana | Male | Owner | 49 | 3 | No (formal) education |
| 23 | Tunde | Male | Owner | 58 | 1 | No (formal) education |
| 24 | Yussif | Male | Owner | 38 | 3 | No (formal) education |

**Fig 2. Food safety capabilities, opportunities, and motivation.**

improvements in understanding microbial contamination sources, healthy chicken management, safe slaughtering practices (defeathering and evisceration), separating food preparation activities, personal hygiene, handwashing, workspace cleanliness, safe handling of vegetables and seasonings, and proper food storage. Vendors reported increased confidence in their ability to protect public health. They recognized their crucial role in selling safe chicken at outlets, safeguarding both consumers and vendors from foodborne diseases: *"Now one thing is, for sure, something that goes into the belly has to be clean"* (Ibrahim, owner).

*"What I remember is that they said that the vendors are the ones who feed the population the most, given the importance of our activities [. . .]. The trainers were promoting the health of the population. As if they were looking to help the population, according to me. Anyway, I think that the quest for cleanliness is to help the population as well as us, vendors"* (Abubakar, owner).

For many vendors, training served as a reminder or wake-up call to adhere to food safety measures. Despite having a fair understanding already, some recognized their habitual unhygienic practices after seeing demonstrations of "good and bad" practices through videos and photos during training. They attributed these practices to ignorance or laziness over the years.

*"It is like when someone is sleeping, and we'd wake him up suddenly; it's because he was sleeping in a bad position. We would wake him up so he could change his posture"* (Tunde, owner).

*Education, work experience, and self-efficacy.* Psychological capabilities impacting food safety behaviors also included educational attainment and work experience. Vendors with limited education in particular valued training, while others felt their existing level of knowledge sufficed due to their educational background. Outlet owners highlighted the successful combination of their work experience, spanning from one year to over 40 years in the industry, with training-acquired knowledge. Self-efficacy, vendors' belief in their ability to adhere to food safety measures, was indirectly mentioned often. For many outlet owners, especially those lacking formal education, the training certificate served as a confidence booster, affirming their capability to operate their businesses safely and avoid mistakes that could harm consumer safety. For some, the training renewed a sense of ownership: *"Food safety, in my opinion, is mainly a matter of taking ownership, making sure that you don't get sick after preparing and eating the chicken" (Issa, employee).*

Lastly, particularly outlet owners appreciated the module on business skills.

**Physical capabilities.** *Tool applicability and usefulness*. Most vendors found that learning how to correctly use equipment provided, such as cutting boards, table coversheets, aprons, garbage bins, and handwashing devices with soap, highly useful. This enabled them to directly improve food safety behaviors and improved risk detection visually. For instance, the coversheet used to cover the food preparation table, made it easier for vendors to detect dirt and prompt action: *"The bag we use to cover the table also makes a difference in terms of cleanliness because when you put a utensil down, after two hours you'll see dust settling on it"* (Abubakar, owner).

## Opportunities

**Physical opportunities.** *Food safety equipment*. Almost all vendors considered receiving the equipment as the most significant change, enabling them to implement their gained knowledge and safely prepare chicken meat at their outlets. They found the equipment highly relevant, practical, and easy-to-use in their daily operations:

*"The biggest change is the fact that we were given working materials so that we could strengthen our activities. This is part of the changes since there were no (materials) before. But as soon as they were given to us, there was a change at that level" (*Ahmed, owner).

*Financial resources.* Participants identified financial resources being key for implementing food safety behaviors. Despite strengthened intentions, almost all vendors mentioned a lack of money hindering their efforts. Financial resources were needed for renovating or rearranging outlets, purchasing equipment, and meeting regulatory requirements like health cards for employees: *"What the training requires us to put into practice in the development of our working environment is difficult because of the lack of our financial resources"* (Alhaji, employee).

*Cost of living and uncertainties.* Additionally, limited financial resources were often linked to a stagnant market, high inflation, and rising commodity prices. Consequently, some outlet owners had to reduce salaries, lay off employees, and prioritize funding their children's education or supporting their families instead.

*"[It] is a question of means. It's like what I said before. If you don't have money... because everything has become a matter of means...If you don't have money, you can't do a big project. Even for a bicycle, if you don't have money, you can't have one. You can walk to work, but you thank God that you can come to work and provide for your family, because that's as far as it goes"* (Tunde, owner).

According to vendors, in some cases the prices of live chicken increased (up to 3750 FCFA, equivalent to 6.1 USD (Exchange rate 1 USD-599.9 FCFA, December 2022). Consequently, some experienced a decrease in sales and customers, resulting in their inability to invest in their businesses. Combined with strong competition, vendors reported that high live chicken prices hindered them from buying good-quality and healthy (safe) chickens, increasing the risk of selling unsafe food: *"You used to be able to get 1000 FCFA profit (equivalent to 1.6 USD) on a chicken. But with chickens these days, some people are struggling to make 250 FCFA profit (equivalent to 0.4 USD)"* (Yussif, owner).

Related to the increased costs of living were both owners' and employee's concerns about the unstable political situation in Ouagadougou at the time of the interviews, and other societal challenges which resulted in a lower prioritization of food safety:

*"Our work now is no longer work. [...] Before, we knew we were working, but now there is no work because you can come and sit all day until night without even getting five FCFA francs. But before, if I wasn't there at seven o'clock, people would call me. But now everyone sees the current situation in the country. Everyone is looking for peace, so the food side is no longer as urgent as the soul wants it to be, because now there are many problems"* (Tunde, owner).

Despite challenges, many outlet owners invested in new equipment like liquid soap and cleaning materials, replaced hardware, and had renovated outlets with tiles, windows, or repainted walls. Some received financial support from friends or family through small loans. Accessing these loans was seen by vendors to invest in food safety without losing business. Despite increased expenses, vendors perceived these investments positively, noting improved client satisfaction and attraction.

*Market infrastructure.* Improving market infrastructure and city cleanliness were seen as opportunities. However, limited operating space at market sites posed barriers, hindering vendors from installing sufficient materials or separating activities to ensure food safety and chicken dish safety. One owner mentioned challenges from a nearby mechanic's workshop

affecting hygiene and waste flow into their outlet. Others shared space with drinking establishments, creating additional cleanliness challenges. Poor environmental hygiene and roadside dust posed major challenges to keeping outlets and food clean. Outlet owners created provisional barricades like windows or covered the workspace to prevent dust from settling on food and equipment: *"The dust there, if you clean it in the morning, it's dirty again by noon"* (Jacques, owner). One vendor noted sun exposure and heat necessitated roofing over the grilling area, posing financial and cleanliness challenges. Other issues included inadequate rainwater drainage, eateries situated near drains, and a lack of nearby waste disposal systems:

> *"What we tried in vain was in relation to the draining ditch because . . .we had a lot of discussions on that because many people had said that it is not normal that we place ourselves on the gutters. But in Ouagadougou, if we take about 100 vendors, those who are not on the draining ditches, well, it will not exceed five. [. . .] It's first because we don't have the choice, so if you look at where people are concentrated. . . There are a lot of slaughter slabs [. . .] If we look over there, some of the draining ditches collapsed. These are the things that we are worried about"* (Mohammed, employee).

*Regulatory environment*. Many vendors highlighted regulatory challenges, including the complexity and delays in obtaining employee health cards (see Box 1). The varying costs from town hall officials, agents, or middlemen, and the issuance of cards without medical checks and vaccinations, raised doubts about their usefulness among vendors.

> *"Concerning the hygiene license, I applied for it last month, but the papers are still not ready. Often, they keep papers for a long time, but the police will chase you, they will summon you and then you can't explain yourself. I applied even though the paper had not completely expired, but so far, no news [. . .] We must put pressure on those in the offices to make things happen, otherwise it's not easy"* (Mohammed, employee).

---

### Box 1. Health cards

Law No. 23/94/ADP on the Public Health Code states in Article 36 that **any person working in an establishment manufacturing and selling foodstuffs must be subject to health control, prevention and treatment measures**. As part of the follow-up, a professional health card is issued. The authority determines the required exams to obtain this card. In Ouagadougou, there are three examinations including hepatitis B, stool and urine. The card includes a stamp of 1,000 FCFA and two identity photos. The validity of the card is 8 years with annual renewal of exams. A fine of 16,000 FCFA is imposed on people who do not have an updated professional health card.

---

*Continued training*. Vendors expressed that they never stopped learning and believed food safety trainings to be vital: *"There is no such thing as advice fatigue"* (Ahmed, owner). During interviews, most vendors emphasized the need for regular follow-up or refresher trainings and advancements in training. Some suggested follow-up sessions to reinforce lessons learned and prevent reverting to old practices, while others looked forward to new techniques for enhancing food safety in the future.

## Social opportunities

*Social support, spillovers, and approvals.* Gaining support from colleagues and neighboring vendors was crucial and seen as a key social opportunity to collectively enhance food safety within the market community. Some vendors shared their lessons directly with others, potentially creating spill-over effects, while others indirectly encouraged their colleagues to maintain hygiene practices.

> *"As far as cleanliness is concerned, I really put a lot of effort into it, and my neighbors can testify to this, because when I clean my sales outlet, and if they are not there yet, I also clean their space. Because if I sweep my point and theirs is dirty, the wind can blow the rubbish onto my space"* (Abubakar, owner).

Employees emphasized the crucial role of support from outlet owners and managers in implementing lessons learned. Financial approvals and trust in employees' capabilities to improve safety ultimately rested with owners. Some employees therefore had to negotiate changes or faced limited decision-making power: *"It is difficult as an employee to say exactly what changes need to be made because the decision and the means to do it rest with the boss"* (Sherif, employee).

Some vendors expressed concern over insufficient interest or support from food safety regulators and town hall officials. They emphasized the need for inspectors to be actively involved in training and supporting vendors, rather than penalizing:

> *"At the level of the agents of the town hall: they must review their way of doing things. They should try to be on our side, and not summon us all the time. Sometimes we acknowledge that we are in breach of the law, but sometimes we don't realize the seriousness of the error when we are checked. It gives the impression that they just go out to fine us. Instead, we want them to supervise us"* (Hakim, owner).

*Outlet culture.* Vendors noted that outlet culture and the attitudes of untrained employees were important factors. Older vendors mentioned that younger employees' stubbornness or lack of interest hindered change, occasionally leading to clashes over best practices:

> *"I almost got into a fight with one of the employees there. He went to serve someone, and when he served the person, he came back and didn't wash his hands. Then someone ordered chicken, and he went to cut up the chicken. So, I told him to wash his hands first and then he told me 'I work here, I know how to do my job, but that I should not show him how to do the work, as I don't have the right to say that, because they have been here longer than me. I'm not here every day. But the day I'm here I want to show them what they must do, and we almost had a fight because of that. [. . .] It's because they weren't trained"* (Amina, employee).

One owner hired additional employees to ensure cleanliness. Sensitizing and training other employees to maintain hygiene standards was thus seen as a key opportunity to enhance food safety, with constant supervision believed to be essential.

*Employee turnover.* Outlet owners were concerned about high employee turnover and unreliability (e.g., theft), which hindered change implementation. They noted time lost in training new employees who would soon leave, and the financial risk of purchasing health cards due to frequent resignations and dismissals.

*"The individual health record is a problem because you can recruit an employee and then you can get rid of them in a short period of time. In this situation when you recruit another one and the controllers come, they will ask for the new employee's book. That's what really bothers us, in one hour you can hire and fire. We can pay but the employees don't stay. It is a loss"* (Sulemana, owner).

## Motivation

**Reflective motivation.** *Perceived sense of responsibility*. After training, vendors were mostly motivated by a heightened sense of responsibility for consumer and vendor health, driven by an increased ability to prevent the sale of unsafe chicken and foodborne diseases.

*"It's the dirt that can lead to diseases so the risks will not be limited to the worker only but all the customers, heh! So, it's an obligation for the vendor to secure the cleanliness of the space and the food. Food safety isn't just a good thing. It's an obligation"* (Roland, owner).

*Reputation, client attraction, profits, and competition*. Many vendors connected their sense of responsibility and adherence to food safety with preserving their reputation, attracting, and retaining clients. They understood that consumers were "not interested" in getting sick from unsafe chicken and that their investments could enhance client satisfaction and profits. Not adhering to food safety measures would risk them losing market position and driving away customers, especially with competition from neighboring vendors. Post-implementation, vendors saw returning customers and received compliments for equipment upgrades and improved layouts:

*"When I replaced the wood block with the cutting board they gave us, customers always enquired where I got it from. They really like it. As soon as someone arrives, they quickly ask where I got it from"* (Yussif, owner).

Aside from direct food safety improvements like cleanliness, some owners believed that making the workplace more attractive with repainting or new equipment attracted more clients. This increased their commitment to continue implementing changes and expand their business. In addition to these perceived benefits, owners felt they slowly started to financially benefit from the training. Some noticed orders increased and realized that their investments would ultimately yield more financial stability. It was also realized that unsafe practices, such as selling sick chicken though profitable for a short time, could lead to bankruptcy in the longer term. Interestingly, retaining high-value customers who demanded luxury and higher cleanliness standards motivated one owner to differentiate his outlet: *"If people from high categories stop to buy chicken in a dirty setting it is really not suitable for their social category"* (Sulemana, owner).

*Client satisfaction*. Most vendors prioritized client satisfaction and balanced this with hygiene measures. Some noted that customers explicitly requested for safe chicken, while other customers disapproved certain safety practices such as plucking in hot water or using gloves, fearing it would change taste. Some customers also reminded vendors of hygienic standards, encouraging better behavior.

*"As we want the market, we respect their wishes. If they think the water is dirty, for example, we change the water and try to follow their wishes. Others get angry or even insulting and are ready to leave. I often find myself apologizing, negotiating with them so as not to lose the*

*market or to get a bad image. That's why I don't hesitate to follow their wishes regarding hygiene"* (Hakim, owner).

*Consumer demands and time pressure*. Customers' knowledge motivated vendors to enforce safe practices, with some customers overseeing every step from selecting live chicken to grilling and seasoning. Time pressure, however, posed a significant challenge and often led to conflicts, as some clients preferred fast food over safe food. Safe chicken preparation increased time, and vendors sometimes compromised safety to satisfy impatient clients and avoid losing business:

*"It's the behavior of some people who are in a hurry. For example, some of our former customers. . ., he identifies the chicken he wants, and he tells you to serve it like that. [. . .] When we tell them not to touch the food directly or to put it on the cutting board, they say to do it quickly or they will leave"* (Ibrahim, owner).

Other challenges related to consumer demands were the fear of customers that the taste of the chicken would change after changing certain practices:

*"At the training level we were told to abandon this way of doing things and pluck the chicken in hot water, wash it to get rid of the dirt before grilling. However, by doing so the taste is not the same. People have accepted this but are complaining. We sensitize them by saying that the way we did it before was dirty and we recognize that while our current way is not as good as the old one, it is clean"* (Roland, owner).

*Regulatory compliance*. Finally, outlet owners emphasized the importance of complying with food safety regulations. Unannounced inspections by town hall officials kept them vigilant about presenting clean outlets, because of risks of fines and sanctions. *Ha*! *They don't even warn us before they arrive for their inspection*! *I would like them to be happy even on the day they surprise me. We need the health cards to present in case of control.* (Ibrahim, owner).

Ownership of the right documents also increased vendors' ability to gain more profits, serving as an extra motivator to adhere to food safety measures.

**Automatic motivation.** *Passion*, *habits*, *faith*, *and fear*. Although mentioned to only some extent during interviews, automatic motivations were expressed in the form of feeling passionate about work, morality and beliefs, ingrained habits and routines, as well as emotional reactions or feelings tied to food safety. One vendor felt that selling safe chicken ultimately, was beyond his control and left this up to faith. It was however mostly vendors' passion that motivated behavior to change. *"I say to myself, if you are passionate about your work, you take advantage of every opportunity to ensure that your work also advances a little. And what they showed us there was really important"* (Mohammed, employee).

A lack of passion or carelessness and unwillingness to invest further in practices or business were seen as detrimental to food safety and business:

*"If I abandon the hygiene practices, it means that I don't want to work anymore. A Mossi proverb says: 'if you don't want to stop farming, you need to buy a new hoe'"* (Abubakar, owner).

*Reinforcement*. The certificate and equipment served as reminders to sell safe chicken, reinforcing hygienic behaviors. Some vendors struggled, however, to break from unsafe habits or beliefs. For instance, one owner preferred to pluck chicken dry immediately after slaughtering due to taste concerns affecting customer retention, contrary to training to wash it before

evisceration. Others found it challenging to use forks instead of bare hands when handling grilled chicken, believing they lacked skill, slowing down work and causing customer conflicts.

> *"Using the fork is a bit complicated. Not everyone is comfortable with its use. [. . .] I did not say that we are going to abandon the use of the fork, but it will be less. Especially on Saturdays and Sundays with the influx of customers, this slowness can lead to disputes"* (Jean, employee).

## Discussion

This study provided a comprehensive overview of the impact of a food safety intervention aimed at improving food safety practices of vendors selling ready-to-eat chicken in informal markets in Ouagadougou, Burkina Faso. While previous RCT's findings showed that training combined with equipment significantly enhanced vendor food safety knowledge and behavior [30], this qualitative study sheds light on how the intervention specifically influenced vendors' food safety capabilities, opportunities, and motivations, using the COM-B model. Findings highlighted increased perceived capabilities and motivations related to food safety behavior post-training as reported by the vendors, but moreover this study identifies significant opportunities and persistent barriers to implementing behavior changes, despite vendor willingness and intentions to change.

First of all, vendors reported a major boost in their capabilities specifically regarding knowledge gains and awareness, as well as an improvement in their skills to perform food safety behaviors, post-training [30]. In line with RCT findings, knowledge increases were perceived in relation to live chicken management, slaughtering, serving, and seasoning practices. Behavior-based training combining knowledge and behavior modification techniques including how to handle equipment strengthened a sense of responsibility to change behavior and affected vendors' risk perceptions [37]. Vendors' motivation to adopt safe food practices was strongly driven by a sense of moral responsibility for consumer and employee health and the ability to prevent foodborne illness. This intrinsic motivation is crucial for sustainable behavior change, as it stems from the belief in the effectiveness of the behavior to reduce risks (response-efficacy) and confidence in one's ability to perform the behavior (self-efficacy) [38]. Moreover, perceived improvements in both awareness and self-efficacy resulted in a renewed sense of food safety ownership.

Secondly, vendors' motivation to change was driven by increased perceived benefits related to client satisfaction, attracting and retaining clientele, as well as the perceived obligation to adhere to hygiene regulations. Outlet owners seemed especially motivated by the perceived increased profits associated with improved hygiene, through higher sales and attracting new consumers, and the prospect of financial stability in the longer term. These findings are in line with the RCTs findings that showed reported mean increases in daily profits and clients [21]. However, despite increased motivations, fear of change, and vendors' inability to break habits hindered implementation of changes, whereas passion and perceived importance of continuous learning was seen as motivator to break these habits.

Interviews uncovered additional opportunities and barriers beyond what the RCT findings revealed. For instance, customer criticism towards new practices, and time pressure during peak hours compromised food safety behavior. Henson and colleagues illustrate how vendor motivation is influenced by incentives to adopt food safety practices. Such incentives "reflect the benefits and costs for the enterprise"[10]. The economic and social costs and benefits include the market's regulatory system, social pressure from vendors, and rewards for better

food safety measures driven by consumer demand [38]. With consumers becoming more aware about food safety issues in LMICs [39], they may act as agents of change through demanding safer food as an additional motivator [3]. Thus, empowering consumers to shift norms may be an effective tool for driving the food safety agenda in informal markets. For example, a study assessing the impact of a consumer targeted communication behavior change campaign to improve consumer demand for safer food, campaign in Ouagadougou, showed promising effects: campaign recall was associated with improved consumer knowledge, awareness, and perceived benefits related to buying from outlets that safely handle poultry meat In East-Ethiopia, recall of a similar mass-media campaign targeting consumers was associated with the purchase of safer tomatoes in informal markets [40].

Vendors reported poorer food safety practices during peak hours, aligning with evidence that non-compliance rates are highest during busy times for both motivated and poorly motivated food vendors [41]. This pressure forces vendors to balance client satisfaction with safety measures. Consumer concerns can both reinforce and hinder safe practices. Training that enhances capacity and self-efficacy can ultimately help vendors discern between demands that promote unsafe or safe outcomes [38]. Additionally, outlet culture and social support was seen as essential to improve food safety behavior. However, resistance from untrained employees, high turnover and associated financial risks, and the need for constant supervision perpetuated unsafe practices, according to outlet owners. Training untrained colleagues is therefore essential to avoid employee resistance and the need for constant supervision, an approach that was successful in improving food safety knowledge and practices among meat producers in Nigeria [42].

Distributed equipment was believed to be the most significant driver (and opportunity) for behavior change, which is in line with findings showing significant behavioral improvements in domains where equipment was provided [21]. For instance, handwashing increased with the distribution of handwashing devices, and waste management improved with the introduction of bins. Equipment thus facilitated behavior change by directly enabling or reminding of the implementation of practices, increasing capabilities, and boosting motivation.

After training, owners remained concerned about structural barriers to implementing food safety behaviors: limited financial resources, poor market infrastructure, and regulatory challenges. These measures necessitated investments in equipment and outlet layout, explaining limited changes in behavioral domains requiring financial outlays. Financial constraints were exacerbated by rising living costs, inflation, and higher live chicken prices, prompting owners to deprioritize food safety measures. Limited operating space, dust, heat, and inadequate water and waste disposal systems pose significant challenges. Such market infrastructure deficiencies are recognized in literature as fostering contamination and impeding safe food practices in informal markets [3,10,43]. In Burkina Faso's urban informal markets, low income and poor state of market infrastructure and amenities such as potable water, clean environment, and pest control have been linked to poor hygienic practices among chicken vendors [11]. In Phnom Penh, Cambodia, vegetable vendors had higher perceived motivation and capability to implement food safety practices than perceived opportunities to do so [44]. Similarly, other studies on food vendors' food safety practices in Bangladesh, Vietnam, Ghana, and South-Africa show that infrastructural deficits to street food vending including a lack of operational space and basic facilities are a major shortfall to the application of knowledge and foster attitudinal change [45–48].

Despite recognizing the importance of regulations, many vendors doubted their usefulness due to complexity and distrust in regulators. Barriers included complex health card acquisition processes and lack of support from regulators. These shortcomings in food control oversight in LMICs stem from weaknesses in regulatory frameworks and control systems. In Tanzania,

informal dairy vendors criticized high taxation, bribery, and harassment from officials, despite lacking government support [49]. Studies from Sub-Saharan Africa and South Asia showed that informal market governance involves bribery, fines, harassment, confiscation, violence, forced closures, and threats [1,50]. These counterproductive strategies reduce vendors' incentives to invest in safe food practices and infrastructure [51–54]. The regulatory gaps are underpinned by insufficient human, financial, and technical resources, ultimately hindering effective implementation and highlighting the need for regulatory reforms and increased resources to secure safer food supply chains in LMICs [10,55].

## Methodological considerations

This qualitative study embedding the Most Significant Change technique [35] allowed for a nuanced exploration of trained vendors' experiences through capturing rich narratives and provided a holistic understanding of the intervention's impact as perceived by vendors themselves. Including diverse groups of vendors, both owners and employees with varied socio-demographic backgrounds and experience, enabled us to capture a broad range of perspectives. Despite predominantly male representation, our study population was representative of the ready-to-eat chicken sales landscape. Nevertheless, our study was limited to Ouagadougou's informal markets potentially restricting the transferability of findings to other contexts. The study design did not allow for discussing stories of change with other key stakeholders, such as inspectors. Future studies could engage these actors in impact evaluations to enhance data scope. Interviews conducted shortly after training minimized recall issues, ensuring data accuracy and reliability. However, follow-up interviews months later could evaluate long-term training sustainability and capture evolving perspectives and challenges. Despite efforts to foster openness, social desirability bias, especially in addressing behavior changes needing owner approval, may have influenced responses. Finally, although the COM-B model provided a broader understanding of food safety behavior changes post-training, it may be less suitable for identifying psychological determinants and pathways to behavior change.

## Conclusion and implications

This study showed the importance of increased capabilities and motivations when seeking to improve food safety behaviors. Behavior change goes beyond knowledge acquisition and depends on risk perceptions. Findings show how contextual opportunities and barriers related to financial resources, market infrastructure and regulations, influence implementation of food safety behaviors promoted in trainings in informal settings. Investing in infrastructure and engaging vendors in navigating these challenges are therefore crucial for sustainable effects. Collaborative alliances between vendors and regulators, rather than punitive measures, may offer a pathway to addressing regulatory challenges. In the short term, training sessions should explicitly address and engage vendors in finding solutions to navigate the structural, economic, and social barriers they face. Client satisfaction and increased profits, as well as continuous learning, could strategically incentivize behavior change. Additionally, tapping into spillover effects among neighboring vendors, e.g. through peer-to-peer learning and role models, may provide pathways to improve food safety within a retail sector community.

## Supporting information

**S1 Table. Training overview with key messages.**
(DOCX)

**S2 Table. Interview guideline (with main questions).**
(DOCX)

**S3 Table. Coding scheme.**
(DOCX)

**S1 File. Inclusivity in global research questionnaire.**
(DOCX)

## Acknowledgments

We would like to express our gratitude to the research team, in particular Ms. Yameogo and Mr. Diarra, as well as all vendors who were willing to share their stories with us.

## Author Contributions

**Conceptualization:** Dónya S. Madjdian, Marcel van Asseldonk, Theodore J. D. Knight-Jones, Emely de Vet.

**Formal analysis:** Dónya S. Madjdian, Vera Dankwah Badu.

**Funding acquisition:** Theodore J. D. Knight-Jones, Emely de Vet.

**Investigation:** Dónya S. Madjdian, Vera Dankwah Badu, Guy Ilboudo, Valerie R. Lallogo.

**Methodology:** Dónya S. Madjdian, Vera Dankwah Badu, Valerie R. Lallogo, Michel Dione, Marcel van Asseldonk, Theodore J. D. Knight-Jones, Emely de Vet.

**Project administration:** Guy Ilboudo, Valerie R. Lallogo.

**Supervision:** Guy Ilboudo, Valerie R. Lallogo, Michel Dione, Marcel van Asseldonk, Theodore J. D. Knight-Jones, Emely de Vet.

**Validation:** Dónya S. Madjdian, Vera Dankwah Badu, Valerie R. Lallogo.

**Visualization:** Dónya S. Madjdian.

**Writing – original draft:** Dónya S. Madjdian, Vera Dankwah Badu.

**Writing – review & editing:** Vera Dankwah Badu, Guy Ilboudo, Valerie R. Lallogo, Michel Dione, Marcel van Asseldonk, Theodore J. D. Knight-Jones, Emely de Vet.

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
