## [Decision Letter · Decision Letter 0]

24 Sep 2024

PONE-D-24-30028

Fast food over safe food? A qualitative evaluation of a food safety training intervention for street vendors applying the COM-B model in Ouagadougou, Burkina Faso

PLOS ONE

Dear Dr. Madjdian,

Thank you for submitting your manuscript to PLOS ONE. After careful consideration, we feel that it has merit but does not fully meet PLOS ONE’s publication criteria as it currently stands. Therefore, we invite you to submit a revised version of the manuscript that addresses the points raised during the review process.

We look forward to receiving your revised manuscript.

Kind regards,

Md. Tanvir Rahman, DVM, MSc, PhD

Academic Editor

PLOS ONE

Journal Requirements:

4. Thank you for stating the following financial disclosure: “This study was financially supported by the Bill & Melinda Gates Foundation, UK Government Foreign, Commonwealth & Development Office (FCDO) - UK Aid from the United Kingdom government [INV-008430-OPP1195588] and the CGIAR Research Program on Agriculture for Nutrition and Health.”

5. We note that you have indicated that there are restrictions to data sharing for this study. PLOS only allows data to be available upon request if there are legal or ethical restrictions on sharing data publicly. For more information on unacceptable data access restrictions, please see http://journals.plos.org/plosone/s/data-availability#loc-unacceptable-data-access-restrictions. Before we proceed with your manuscript, please address the following prompts: a) If there are ethical or legal restrictions on sharing a de-identified data set, please explain them in detail (e.g., data contain potentially identifying or sensitive patient information, data are owned by a third-party organization, etc.) and who has imposed them (e.g., a Research Ethics Committee or Institutional Review Board, etc.). Please also provide contact information for a data access committee, ethics committee, or other institutional body to which data requests may be sent. b) If there are no restrictions, please upload the minimal anonymized data set necessary to replicate your study findings to a stable, public repository and provide us with the relevant URLs, DOIs, or accession numbers. For a list of recommended repositories, please see https://journals.plos.org/plosone/s/recommended-repositories. You also have the option of uploading the data as Supporting Information files, but we would recommend depositing data directly to a data repository if possible. We will update your Data Availability statement on your behalf to reflect the information you provide.

6. Please include a separate caption for each figure in your manuscript.

Additional Editor Comments:

Dear Authoirs,

Please update the manuscript as suggested by the reviewer.

Best wishes,

Prof. Rahman

Academic Editor

==

Professor Md. Tanvir Rahman

DVM, MSc (Canada), PhD (UK), Postdoc (Germany)

Elected Fellow at Bangladesh Academy of Sciences

Department of Microbiology and Hygiene

Faculty of Veterinary Science

Bangladesh Agricultural University, Mymensingh-2202, Bangladesh.

Tel. + 88-091-67401-6/ Ext. 63215 (Office)

Mobile. +88-01913323307; Fax. +88-02996661510

Ex-Director, International Desk, BAU

Ex-Director, Professor Muhammed Hussain Central Laboratory, BAU

E.mail: tanvirahman@bau.edu.bd

https://vmh.bau.edu.bd/profile/VMH1005

Adjunct Visiting Professor, Xinxiang University

https://www.sites.google.com/site/tanvirahman/Home

Reviewers' comments:

Reviewer's Responses to Questions

**Comments to the Author**

1. Is the manuscript technically sound, and do the data support the conclusions?

Reviewer #1: Yes

Reviewer #2: Yes

2. Has the statistical analysis been performed appropriately and rigorously? 

Reviewer #1: N/A

Reviewer #2: N/A

3. Have the authors made all data underlying the findings in their manuscript fully available?

Reviewer #1: Yes

Reviewer #2: Yes

4. Is the manuscript presented in an intelligible fashion and written in standard English?

Reviewer #1: Yes

Reviewer #2: Yes

5. Review Comments to the Author

Reviewer #1: The authors have conducted commendable work addressing the issue of food safety in chicken sales outlets in Ouagadougou, Burkina Faso. Through a qualitative study, they demonstrate the positive impact of training provided to vendors on improving their food safety practices. The results clearly indicate that the training was beneficial, leading to a significant improvement in vendors' hygiene practices. However, since this is an interventional study, it would have been helpful to include, in the introduction section, results from an assessment of the vendors' knowledge, attitudes, and practices prior to the training. This would offer readers a clearer understanding of the intervention's (training) impact on the improvement of food safety practices among vendors.

I recommend publication after addressing the following minor corrections:

1. Harmonize the referencing system in the text [line 70 (1), (10) and line 78 (14,15)].

2. Remove the description of the COM-B model from the introduction and move it to the methodology section.

3. Add a column to Table 1 to include the order number.

4. Lines 228 to 233: I suggest replacing this verbatim with a testimonial illustrating the impact of psychological skills on improving food safety.

5. Line 267: Also provide prices in US dollars, and similarly for lines 271 and 272.

Reviewer #2: Reviewer comment

The paper entitles “Fast food over safe food? A qualitative evaluation of a food safety training intervention for street vendors applying the COM-B model in Ouagadougou, Burkina Faso “is a really pleasant document to read. It deals with a qualitative analysis of the impact of training on the behavior of chicken sellers as well as owners of outlets, for a better awareness of the harmful effects that a lack of hygiene could cause.

This is a very important topic cause; luck of food safety can negatively impact public health specially concerning ready-to-eat food. More importantly, after the training, evidence can be observed in the field and the testimonies of vendors and owners justify the need to repeat such kind of training.

The scientific way (ethical authorization and vendors and owner consent) the topic is treated involve COM-B methods and is conducted with a good manner and with evidences.

Therefore, I recommend it publication with one small recommendation that does not affect the paper quality. It should be interesting to emphasis that selling ready-to-eat chicken is and activity devoted to men.

6. PLOS authors have the option to publish the peer review history of their article (what does this mean?). If published, this will include your full peer review and any attached files.

Reviewer #1: No

Reviewer #2: **Yes: **COMPAORE Kiswendsida Abdou Muller

---

## [Author Response · Author response to Decision Letter 0]

27 Oct 2024

PONE-D-24-30028 | Revision 1 | Rebuttal Letter

Fast food over safe food? A qualitative evaluation of a food safety training intervention for street vendors applying the COM-B model in Ouagadougou, Burkina Faso

Point-to-point response to the editor’s and reviewers’ comments (in italic)

***

Reviewer #1: 

Comment 0: The authors have conducted commendable work addressing the issue of food safety in chicken sales outlets in Ouagadougou, Burkina Faso. Through a qualitative study, they demonstrate the positive impact of training provided to vendors on improving their food safety practices. The results clearly indicate that the training was beneficial, leading to a significant improvement in vendors' hygiene practices. However, since this is an interventional study, it would have been helpful to include, in the introduction section, results from an assessment of the vendors' knowledge, attitudes, and practices prior to the training. This would offer readers a clearer understanding of the intervention's (training) impact on the improvement of food safety practices among vendors. I recommend publication after addressing the following minor corrections:

Response: We greatly appreciate the reviewer's time and kind compliments on our manuscript. We agree that an overview of the assessment of vendor’s knowledge, attitudes and practices prior to the training strengthened the understanding of the training’s impact on food safety practices. Please see the phrases added to the introduction of the manuscript, on lines: 104-122. We have addressed your valuable corrections in detail below and hope this addressed your comment. 

Comment 1. Harmonize the referencing system in the text [line 70 (1), (10) and line 78 (14,15)].

Response: Thank you for capturing these errors. We have now carefully checked our references and ensured they are harmonized.

Comment 2. Remove the description of the COM-B model from the introduction and move it to the methodology section.

Response: We agree with the reviewer that the description of the COM-B model better fits in the methodology section. We have therefore moved down this paragraph to the beginning of the Methods section, see lines 129-144.

Comment 3. Add a column to Table 1 to include the order number.

Response: Thank you for this suggestion. We have now added a new column to table 1 which includes the order numbers for clarity.

Comment 4. Lines 228 to 233: I suggest replacing this verbatim with a testimonial illustrating the impact of psychological skills on improving food safety.

Response: Thank you for the suggestion to replace the verbatim on the impact on financial / business skills with a testimonial illustrating the impact of psychological capabilities on food safety improvements. We have replaced this quote with the following testimonial: Food safety, in my opinion, is mainly a matter of taking ownership, making sure that you don't get sick after preparing and eating the chicken” (Issa, employee) Please see lines 265-267.

Comment 5. Line 267: Also provide prices in US dollars, and similarly for lines 271 and 272.

Response: Thank you for this important comment. We have converted FCFA into US dollars on lines 314, 319 and 320, using the December 2022 exchange rate (added in between brackets on page 12).

Reviewer 2

Comment 1. The paper entitles “Fast food over safe food? A qualitative evaluation of a food safety training intervention for street vendors applying the COM-B model in Ouagadougou, Burkina Faso “is a really pleasant document to read. It deals with a qualitative analysis of the impact of training on the behavior of chicken sellers as well as owners of outlets, for a better awareness of the harmful effects that a lack of hygiene could cause. This is a very important topic cause; lack of food safety can negatively impact public health specially concerning ready-to-eat food. More importantly, after the training, evidence can be observed in the field and the testimonies of vendors and owners justify the need to repeat such kind of training. The scientific way (ethical authorization and vendors and owner consent) the topic is treated involve COM-B methods and is conducted with a good manner and with evidences. Therefore, I recommend it publication with one small recommendation that does not affect the paper quality. It should be interesting to emphasis that selling ready-to-eat chicken is and activity devoted to men.

Response. We are grateful for the positive feedback of reviewer 2 on our manuscript. Thank you very much for highlighting the important observation that selling ready-to-eat chicken is mainly an activity in which men are involved. We have now emphasized that this is predominantly a men’s activity in this context, see line 150, and backed this up with the following reference: 

Gemeda BA, Dione M, Ilboudo G, Assefa A, Lallogo V, Grace D, Knight-Jones TJ. Food safety and hygiene knowledge, attitudes and practices in street restaurants selling chicken in Ouagadougou, Burkina Faso. Frontiers in Sustainable Food Systems. 2024;8:1448127.

---

## [Editor Report · Decision Letter 1]

29 Oct 2024

Fast food over safe food? A qualitative evaluation of a food safety training intervention for street vendors applying the COM-B model in Ouagadougou, Burkina Faso

PONE-D-24-30028R1

Dear Dr. Madjdian,

We’re pleased to inform you that your manuscript has been judged scientifically suitable for publication and will be formally accepted for publication once it meets all outstanding technical requirements.

Kind regards,

Md. Tanvir Rahman, DVM, MSc, PhD

Academic Editor

PLOS ONE

Additional Editor Comments (optional):

Thanks for addressing all the queries of the reviewers.
---

## [Editor Report · Acceptance letter]

12 Nov 2024

PONE-D-24-30028R1 

PLOS ONE

Dear Dr. Madjdian, 

I'm pleased to inform you that your manuscript has been deemed suitable for publication in PLOS ONE. Congratulations! Your manuscript is now being handed over to our production team.

Kind regards, 

on behalf of

Professor Md. Tanvir Rahman 

Academic Editor

PLOS ONE